# Numerical Investigation of Gas Dynamic Foil Bearings Conjugated with Contact Friction by Elastic Multi-Leaf Foils

Changbao Yang [1,*], Zhisheng Wang [1], Yuanwei Lyu [2], Jingyang Zhang [2] and Ben Xu [2]

[1] College of Automation Engineering, Nanjing University of Aeronautics and Astronautics, Nanjing 210016, China
[2] College of Astronautics, Nanjing University of Aeronautics and Astronautics, Nanjing 210016, China; lvyuanwei@nuaa.edu.cn (Y.L.); zjy@nuaa.edu.cn (J.Z.)
* Correspondence: yangcb@nuaa.edu.cn

**Abstract:** With increasing demand for load capacity and dynamic responses, the new generation of Gas Dynamic Foil Bearings (GDFB), assembled from preloaded elastic multi-leaf foils, has been put forward. It exhibits steady and dynamic characteristics that are more sensitive to the contact friction produced by foil–foil bearings and the housings of foil bearings, distinguishing its response to rigidity from those of classic bearings. This study aims to employ numerical simulations to explore GDFBs conjugated with contact friction using elastic multi-leaf foils. A numerical method with Bidirectional Fluid–Structure Interactions (BFSIs), which strengthens the steady and dynamic characteristics of GDFBs, has been developed. The results showed that the distribution of steady stiffness was closely associated with the configuration of the foil assembly and the state of contact friction. The maximum steady stiffness occurred near the key blocks and was accompanied by the support of bump foils. The contact friction increased the stiffness of the foil assembly, which was unfavorable to its operational stability, and the GDFB was prone to rigidity. The contact friction hindered the pull-back of the top foils near the negative pressure area. An increase in the rotational speed and eccentricity increased the load capacity while decreasing the operational stability. The maximum pressure of the gas film was inconsistent with the maximum deformation along the circumferential direction because of the configuration of the foil assembly. The contact friction induced energy dissipation, whereas increases in the integrated stiffness inhibited loss. The fixed constraint of the key block increased the dynamic cross-stiffness more than the contact constraint did, which was unfavorable to operational stability. The results obtained in this study are significant because they provide a physical basis for GDFBs and can be used to guide their operational optimization.

**Keywords:** gas dynamic foil bearings; contact friction; elastic multi-leaf foils; steady and dynamic characteristics; numerical simulation





## 1. Introduction

Owing to their simple structure, ultra-high speed, and oil-free operation, Gas Dynamic Foil Bearings (GDFBs) have been widely employed in the rotational machinery, including gas turbines, refrigerators, fans, and auxiliary power units, of the aviation industry [1–3]. The load capacity of a GDFB is dependent on the aerodynamic effect produced by the viscid gas film, where the relative motion between the rotor and stator is produced between the wedges [4]. The latest generation of GDFBs is characterized by the support of preloaded elastic foils, which offer improved carrying capacity, stability, and adaptiveness [5]. There is a strong conjugation between these elastic foils and the contact friction, with the contact, detachment, friction, and deformation of the elastic foils significantly affecting the performance of GDFBs [6–8]. Therefore, it is essential to explore the contact friction of foil–foil and foil–bearing housings, which are distinct from classic flat/bump foil bearings.

Scholars and engineers have clarified that the deformation of elastic foils and their contact friction play a significant role in load capacity [9,10]. In particular, the friction force

takes place near the contacted foil when the relative motion has been exerted. For instance, Heshmat et al. [11–13] performed an analytical investigation of elastohydrodynamics in a compliant foil bearing. The most influential parameters were the bearing arc, load angle, number of pads, and degree of compliance. In contrast to conventional approaches, the solution of the governing hydrodynamic equations for a compressible fluid is coupled with the structural resilience of the bearing surfaces. Owing to both structural and hydrodynamic stiffnesses, the solutions include the values of the bearing stiffness coefficients. It is noted that these studies were based on linear friction; the friction force of the foil–foil bearing and the housing of the foil bearing remained unchanged. However, according to Coulomb's law of friction, linear friction cannot completely reflect the practical operations of a GDFB, resulting in a deviation between the numerical simulation and experimental measurement results, especially when coupled with steady friction (or dry friction) and dynamic friction [14–16]. For example, Lee et al. [17] developed a static structural model that considered friction. Multiple static equilibrium positions were presented for static loads owing to friction, indicating its significant effect on dynamic performance. However, the effect of friction on the minimum film thickness, which determined the bearing load capacity, was negligible. Peng and Carpino [18,19] applied mathematical models and numerical schemes to simulate the hydrodynamic pressure and temperature increase of compliant foil bearings. The compressibility of the gas and the compliance of the bearing surface were considered in the bearing system. A numerical algorithm can predict the practical performance and characteristics of foil bearings under extreme operating conditions. Xu et al. [20] conducted an experimental study on the static stiffness of bump foil bearings with preloads. The variation in the static stiffness with the rotational speed presented high nonlinearity, while the static stiffness increases proportionally with the increase in the elastic modulus of the foil assembly. Based on Coulomb's law, it was found that the occurrence of friction improved the stiffness of GDFB. A calculation formula to solve the variable stiffness problem of bump foils was derived from an experiment on the stiffness of bearing structures [21]. Gu Yongpeng et al. [22,23] used the LuGre dynamic friction model to capture accurate stick–slip states of Gas Foil Bearings (GFBs) that coupled with elastohydrodynamics and nonlinear contact friction. The results showed that Friction had an impact on the linear and nonlinear stability of the GFBs. The energy dissipation induced by the foil structure was highly associated with the stick–slip states, such that the optimal friction coefficient considering its nonlinear stability was larger than the linear stability. Brenkacz L et al. [24] employed an ultra-high-speed camera to measure the vibrations of elastic foils. It was found that the camera provided several vibrant pictures of the deformation of the elastic foils in gas foil bearings. The movement of the foils in the frequency domain can be analyzed and compared through numerical simulation. Overall, there is still limited research on contact friction in GDFBs, and the mechanism of these bearings needs to be further explored.

A GDFB is an ideal research object that is strengthened by the aeroelastic interaction between a gas film and a foil assembly. Several scientists pointed out that the aeroelastic mechanism became more complicated with the increased rotating speeds and decreased rigidity of the foil assembly [25–28]. With an increase in the rotating speed, the critical speed and ultimate load capacity increased, and the deformation of the foil assembly decreased. Other researchers explored the characteristics of GDFBs and the optimum parameter values of the foil assembly. For instance, Fatu and Arghir [29] found that manufacturing errors related to the bump height decreased the stiffness of the foil structure within 20 μm and decreased the stiffness of the foil assembly by 50%. Hou et al. [30] suggested that operational flexibility could be enhanced by increasing the number of layers of protuberant foils. In this way, a larger load capacity of up to 43.9 N at $9.5 \times 10^3$ rpm was obtained. They confirmed a strong fluid–structure interaction between the shearing flow of the air-gap film and elastic foils.

The coupling between the gas film and elastic foil significantly impacts the dynamic characteristics and energy dissipation of GDFBs. Overall, the dynamic stiffness increases

with the bearing number ($\Lambda = 6\mu\omega R^2/(P_a c^2)$) and decreases with increasing bearing compliance, whereas dynamic damping exhibits the opposite trend [31]. Additionally, the stability threshold speed of the bearing increases with the eccentricity ratio. Zywica et al. [32–34] pointed out that nonlinear operational characteristics resulted in the dissipation of excess energy and higher vibration levels. The flexible thin foils of a GDFB can operate self-adaptively and maintain a low vibration level at a high rotational speed. The geometry of these foils and contact friction can change the variations in stiffness that occur due to variations in the load capacity. The rotating system significantly affects the preloads and progressive stiffness. Duan et al. [35] estimated the stiffness coefficient and cubic polynomial of the stiffness coefficient using a measured static load. The radius of the top foil could be used to adjust the pressure of the gas film, and the accuracy of the foil shape could be improved. The load capacity and stiffness of the GDFB were twice as large as those of traditional bump-type Gas Foil Bearings (GFBs). Khamari et al. [36] investigated the sensitivity of the length-to-diameter ratio, eccentricity ratio, bearing number, whirl ratio, and bearing compliance to the dynamic stiffness and damping. The stiffness coefficients increased and the damping coefficients decreased, corresponding to speeds of up to 240 krpm. The sensitivity of the dynamic coefficients was highest for the eccentricity ratio and lowest for the whirl ratio. In summary, the contact friction of elastic foils deeply affected their dynamic stiffness and damping, and this effect needs to be investigated further.

As detailed above, several researchers have focused on the contact friction and aerodynamic–elastic coupling in GDFBs, and their influence on the carrying capacity [37,38]. However, the existing understanding of the contact friction induced by the contact between the foil–foil and foil–bearing housing, especially the understanding of the contact, detachment, friction, and deformation of the elastic foils during operation, is extremely inadequate. The performance of a GDFB is closely related to its eccentricity ratio, rotational speed, and friction coefficient. Therefore, it is necessary to investigate the effect of contact friction and its dependent factors on the performance of GDFBs. This study aims to explore GDFBs conjugated with contact friction using preloaded elastic multi-leaf foils. The roles of the eccentricity ratio, rotational speed, and friction coefficient in the characteristics of contact friction and load capacity were elucidated. The results obtained in this study are significant because they provide a physical basis for GDFBs and can be used to guide their operational optimization.

## 2. Computational Method

### 2.1. Physical Model

Figure 1 shows a schematic of a GDFB with a multi-leaf compliant foil. Five pairs of top and bump foils were assembled to construct a multiwedge clearance to support the rotator. For the top foil, the key block was fixed to the slot of the bearing sleeve. The top foils overlapped with each other, and the bump foils were set to support them. Here, $O_1$ is the center of the bearing sleeve, $O_2$ is the center of the rotator, $\omega$ is the rotational speed, $\varphi$ is the deflection angle, and $\theta$ is the circumferential angle. The eccentricity ratio $\varepsilon$ is equal to $e/c$, where $c = R_2 - R_1$. The parameters used in this study are listed in Table 1. Figure 1c shows the relative slide and contact friction force. Subjected to the pressure from the gas film, the deformation of the top and bump foil induced relative slide. The friction force takes place near the contacted foils in which the relative motion has been impeded. The direction of the friction force is opposite to the relative slide. It is noted that the coupling of the startup, acceleration, and stable operation makes the physical mechanism of GDFB more complicated. This study paid attention to the characteristics of the GDFB with contact friction when it works at constant operational parameters and rotational speed. Therefore, the effect of the stage of startup and acceleration, in which dry friction occurs, was ignored. This study focuses on the characteristic of GDFBs conjugated with contact dynamic friction at constant operating parameters.

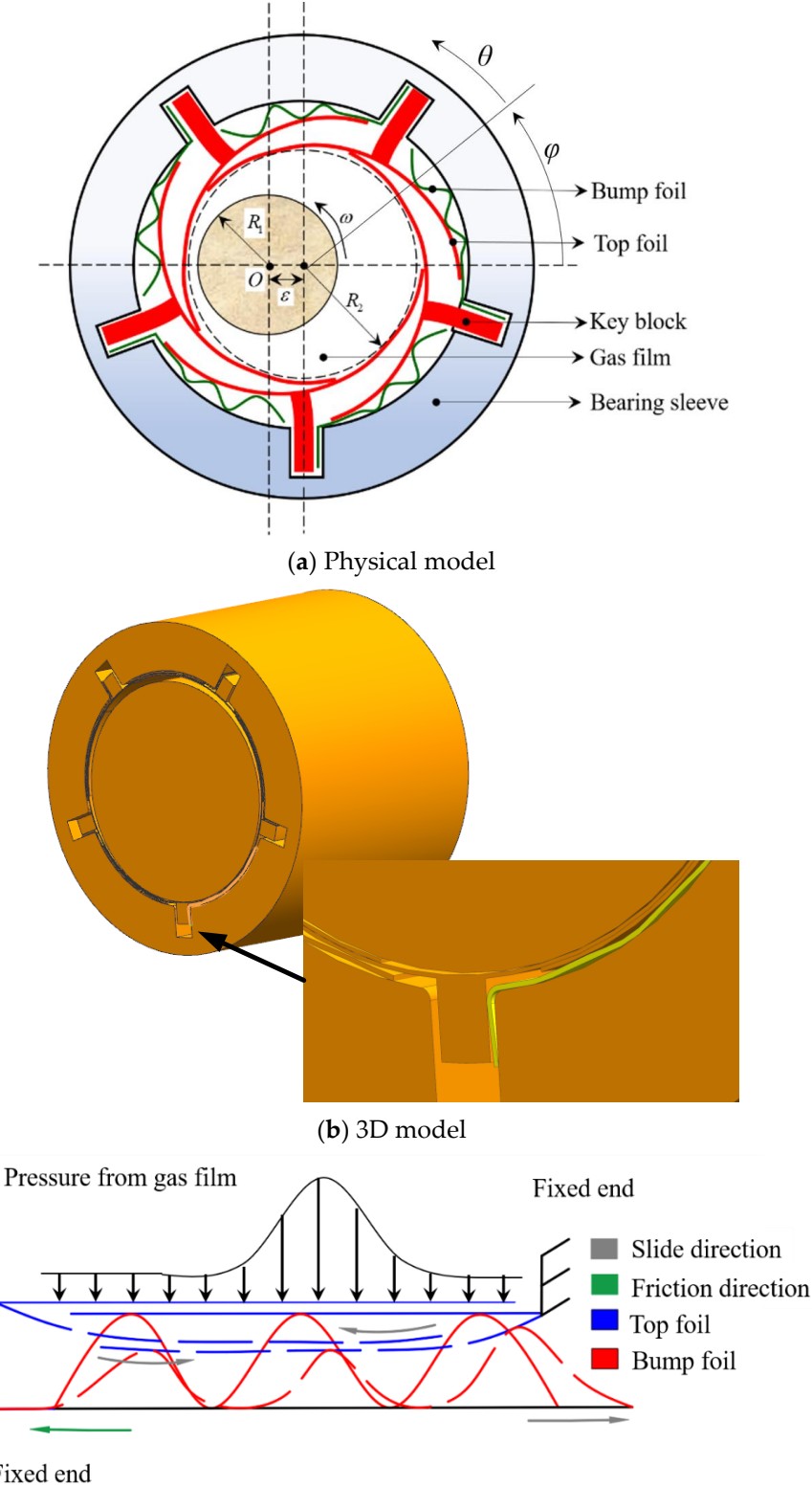

(**a**) Physical model

(**b**) 3D model

(**c**) Relative slide and contact friction

**Figure 1.** Sketch of physical model.

**Table 1.** Parameters employed in this study.

| Parameters | Scale |
|---|---|
| Radius of rotator $R$/mm | 10.95 |
| Number of foils $N_b$ | 5 |
| Radius of stator $R_b$/mm | 11.51 |
| Length of bearing $l$/mm | 25 |
| Thickness of top foil $t_t$/mm | 0.1 |
| Thickness of bump foil $t_b$/mm | 0.13 |
| Number of bumps of bump foil $N_d$ | 3 |
| Elastic modulus of top foil $E_t$/Gpa | 213.7 |
| Elastic modulus of bump foil $E_b$/Gpa | 207 |
| Poisson's ratio of top foil $\nu_t$ | 0.29 |
| Poisson's ratio of bump foil $\nu_b$ | 0.344 |

*2.2. Aerodynamic–Elastic Coupling*

A numerical simulation was conducted on the MATLAB + ABAQUS platform. The pressure and velocity of the shearing flow in the fluid domain were integrated into MATLAB software. The deformation and stress of the elastic foils in the solid domain were solved using the ABAQUS software. Figure 2 shows the aeroelastic coupling procedure. The nominal clearance height $c$ can be calculated from the angle of the top-foil structure angle [39,40]. In this study, the nominal clearance height $c$ was 30.344 µm. In the solid domain, the assembled top foil was subjected to a pressure field induced by shearing flow. It was assumed that the top foil was fully unfolded and did not interact with the bump foil. First, grid division was conducted, and then the initial boundary, which included the elastic foils and the contact friction, was taken into the configuration. The contact was set, and the nonlinear solver was used. Finally, the deformations of the elastic foils were exported. Regarding the computing setting, the top foil, bump foil, and bearing housing were all established using shell elements, and the bearing sleeve was set as a rigid body. Considering the contact friction and damping of foil, transient implicit dynamics were used to analyze the model. The tolerances employed in the iterative calculations for the solid domain was consistent with the fluid domain, which were less than $10^{-5}$ s for the pressure and displacement. Figure 2b shows the data exchange in the aeroelastic coupling. Five steps are shown, as follows: (1) $\bar{h}_0$ and $\varepsilon_0$ are given in the first time-step; (2) drawing aerodynamic force $P_0$ in the fluid domain and then executing it in the interface; (3) processing $\varepsilon^{n+1} = CSD(\bar{h}_n, P_n)$ in the solid domain; (4) executing the displacements of the top foil onto the interface; (5) renovating the grid in the fluid domain and the solid domain. In this study, a steady-state solution in the fluid domain was conducted before the fluid–structure coupling, which was specified as the initial solution.

Using MATLAB 2020 software, based on the initial nominal clearance, the Reynolds equation was used to obtain the film pressure. The pressure data were then computed by the ABAQUS 2019 solver until convergence. Subsequently, the updated film thickness was deduced from the node deformation data of the inner surface of the top foil and sent to MATLAB. After iteration and convergence, the load capacity was obtained by integrating the static pressure of the gas film along the *x*- and *y*-directions.

$$F_x = \int_0^{\frac{L}{R}} \int_0^{2\pi} (p_{i,j} - 1) \sin \theta d\theta d\lambda \tag{1}$$

$$F_y = -\int_0^{\frac{L}{R}} \int_0^{2\pi} (p_{i,j} - 1) \cos \theta d\theta d\lambda \tag{2}$$

$$F_y = \sqrt{(F_x)^2 + (F_y)^2} \tag{3}$$

where $F_x$ and $F_y$ indicate the force along the $x$ and $y$ directions, $p_{i,j}$ is the pressure at node $(i,j)$, $L$ and $R$ are the length and radius of the GDFB, respectively, and $\lambda$ and $\theta$ are along the axial and circumferential directions, respectively.

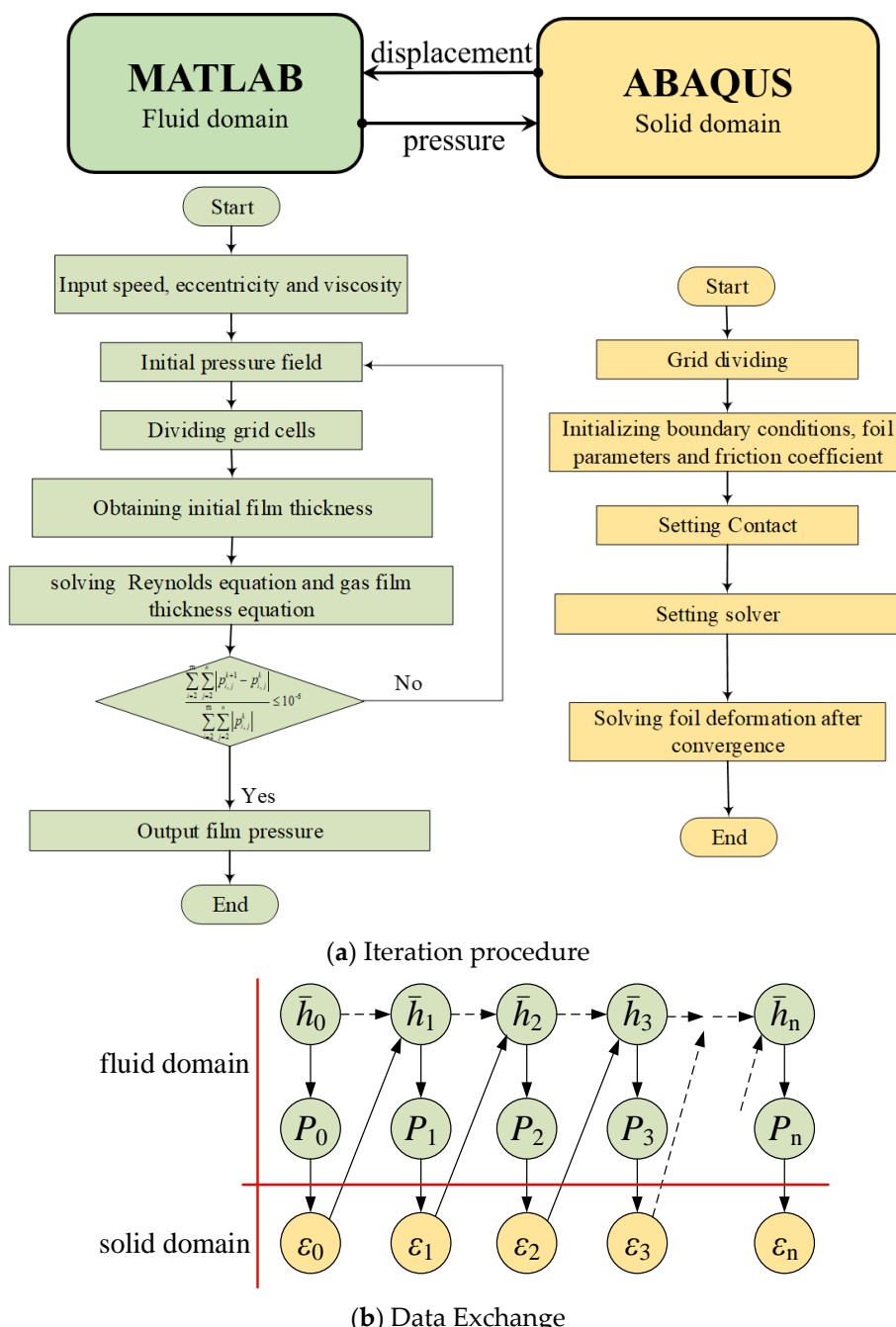

**Figure 2.** Aerodynamic–elastic coupling.

Figure 3 shows the mesh divisions of the elastic foils. In the fluid domain, the cylindrical coordinate system was converted into a plane coordinate system. As shown in Figure 3a, the fluid domain grid was divided as $(0{:}m, 0{:}n)$ evenly along the $x$- and $y$-directions. The end of the $z$-direction was specified as the boundary condition of the pressure outlet, whose temperature and pressure were 300 K and 101,325 Pa, respectively; $x = 0$ and $x = m$ along the circumferential direction coincided, that is, $\overline{P}(0, n) = \overline{P}(m, n)$.

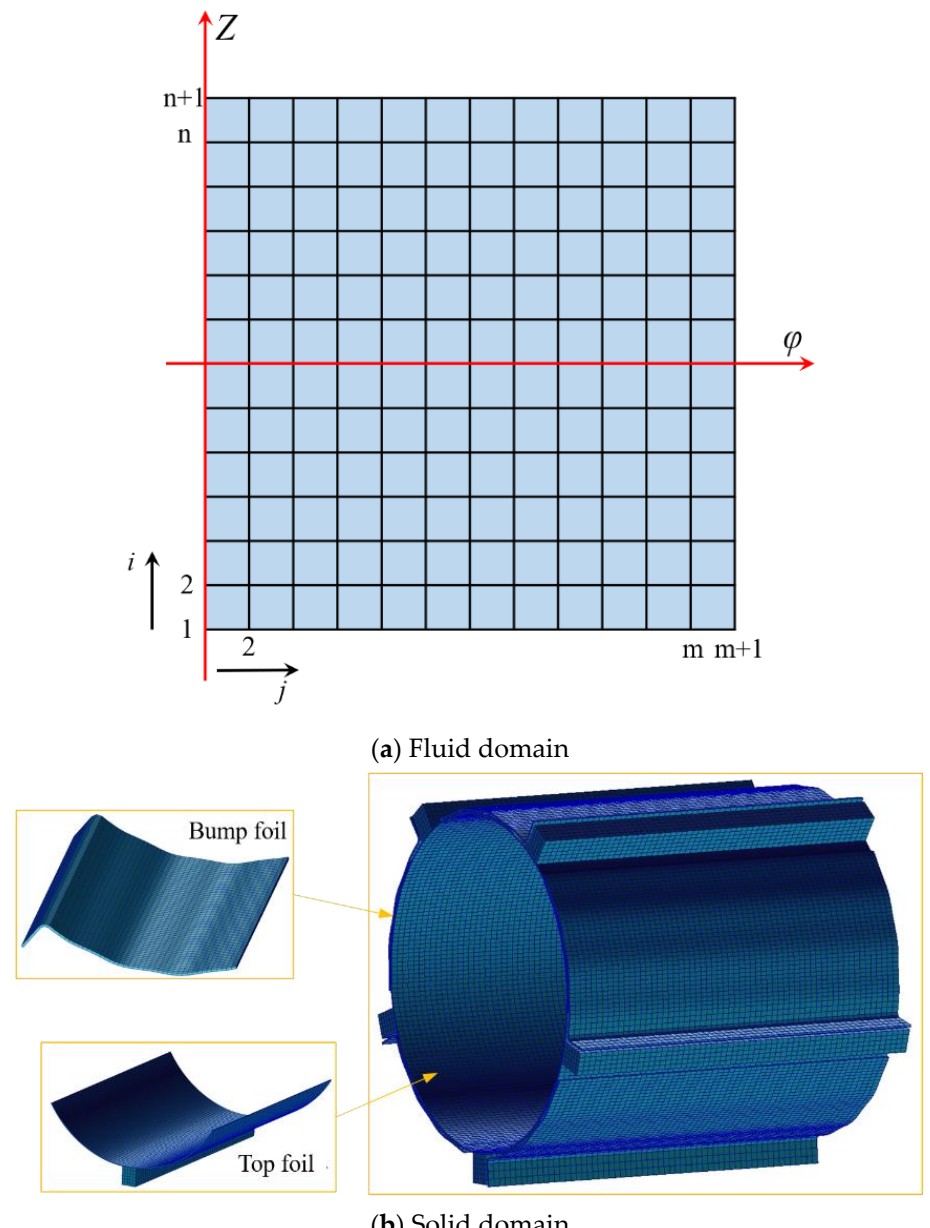

(**a**) Fluid domain

(**b**) Solid domain

**Figure 3.** Grid schematic diagram in the computational domain.

In the solid domain, the elastic foils were divided using a hexahedral grid, as shown in Figure 3b. The top foil was divided by a shell element considering the large deformation of the bump foil in contact with the top foil on the bearing sleeve. The spot weld at the bottom of the top foil was replaced with a constraint set. The boundary conditions in the solid domain were as follows: all the axial degrees of freedom of the foil were constrained and the free end of the bump foil was constrained. The degrees of freedom of the key block along the circumferential direction were constrained by means of assembly with the card slot. Furthermore, the contact friction of the top foil-bump foil bearing, the bearing housing–bump bearing, as well as housing bearing–key block were separately defined. The top foil, bump foil, and key block were specified as deformed bodies, while the bearing housing was specified as the rigid body. The solid domain mesh was specified as a 10-node quadratic tetrahedron with a mesh size of 0.001 mm and a deviation factor of less than 0.1. To reveal the dynamic response of the foil, the grid was structured with a grid type of S4R, and the maximum grid concentration was near the top foil. Solid domain meshes were generated by ABAQUS pretreatment applications. To fully verify the independence of the mesh, the

mesh of the top and support foil was calculated separately. In a grid independency test, the total mesh cell numbers of 1,500,000 were demonstrated to be adequate, such that the computational results were affected by the grid quantity.

### 2.3. Realization of Contact Friction

Figure 4 presents the contact analysis procedure for the elastic foils. The region where the contact occurred kept changing continuously owing to the pressure variation along the circumferential direction near the minimum clearance height. The contact states included sliding, adhesion, and separation. The magnitude of the frictional force was associated with the contact state and region. Based on Coulomb's law, the time step can be expressed as:

$$t_T = -\mu p_N \frac{g_T}{|g_T|} \quad if \; |g_T| > 0 \tag{4}$$

$$|g_T| = 0 \quad if \; |t_T| < \mu p_N \tag{5}$$

where $t_T$ is the tangential friction force; $\mu$ is the friction coefficient between the contact pairs, which depends on the surface properties of the elastic foils; $p_N$ is the normal contact pressure; and $|g_T|$ is the relative sliding speed. When the contact pairs were separated, there was no frictional force. If the tangential friction was less than $\mu_{pN}$ when the contact pairs came into contact, there was no relative sliding; if the relative sliding speed was greater than 0, the tangential friction reached the maximum $\mu_{pN}$ value. The Newton–Raphson incremental iterative algorithm was employed to solve the nonlinear contact friction problem.

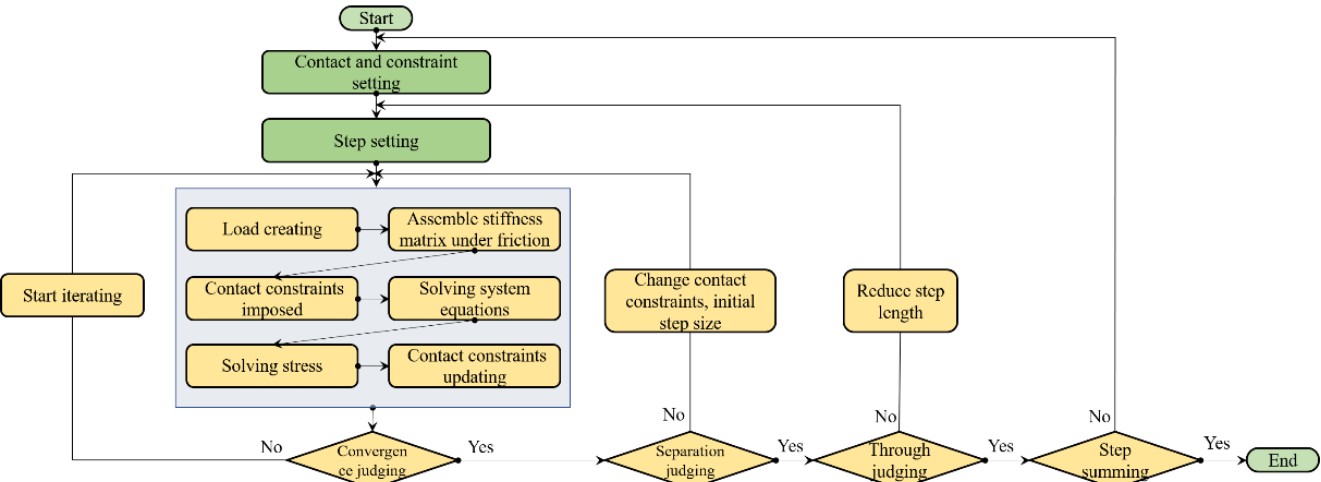

**Figure 4.** Procedure of contact analysis among elastic foils.

### 2.4. Mathematical Description

The procedure for solving the dynamic characteristics is presented in Figure 5. In the fluid domain, the shearing flow in the gas film can be characterized by the dimensionless Reynolds equation and gas film thickness equation based on an isothermal ideal gas:

$$\frac{\partial}{\partial \theta}(\overline{p}\overline{h}^3 \frac{\partial \overline{p}}{\partial \theta}) + \frac{\partial}{\partial \lambda}(\overline{p}\overline{h}^3 \frac{\partial \overline{p}}{\partial \lambda}) = \Lambda \frac{\partial}{\partial \theta}(\overline{p}\overline{h}) + 2\Lambda \frac{\partial(\overline{p}\overline{h})}{\partial \overline{t}} \tag{6}$$

$$h = \overline{G}_0(i,j) + \overline{Z}(i,j) + \varepsilon \cos \theta \tag{7}$$

The corresponding dimensionless parameters include the angle $\theta = x/R$, axial length $\lambda = z/R$, pressure $p = p/P_a$, clearance height $H = h/c$, bearing number ($\Lambda = 6\mu\omega R^2/(P_a c^2)$), and time $\overline{t}(\overline{t} = vt)$, where $\omega$ is the rotational speed, $\Lambda$ is the bearing number, $h$ is the initial thickness of the gas film, $\varepsilon$ is the journal eccentricity, $\theta$ represents the angular coordinates, $h_z$

is the radial deformation of the inner surface of the top foil, $\overline{Z}(i,j)$ is the radial dimensionless displacement of the foils, and $\overline{G}_0(i,j)$ is the dimensionless distance between the top foil and rotator. The dimensionless stiffness and simplified damping of the elastic foils are expressed as:

$$\overline{K} = \frac{Kc}{P_a R^2} \tag{8}$$

$$B = \frac{\eta K_n}{v} \tag{9}$$

where $R$ is the journal radius, $P_a$ is the environmental pressure, $c$ is the nominal bearing clearance, $K$ is the nominal static stiffness, $\eta$ is the coefficient of friction, $K_n$ is the static stiffness of $K(i,j)$, and $v$ is the disturbance frequency of the journal bearing.

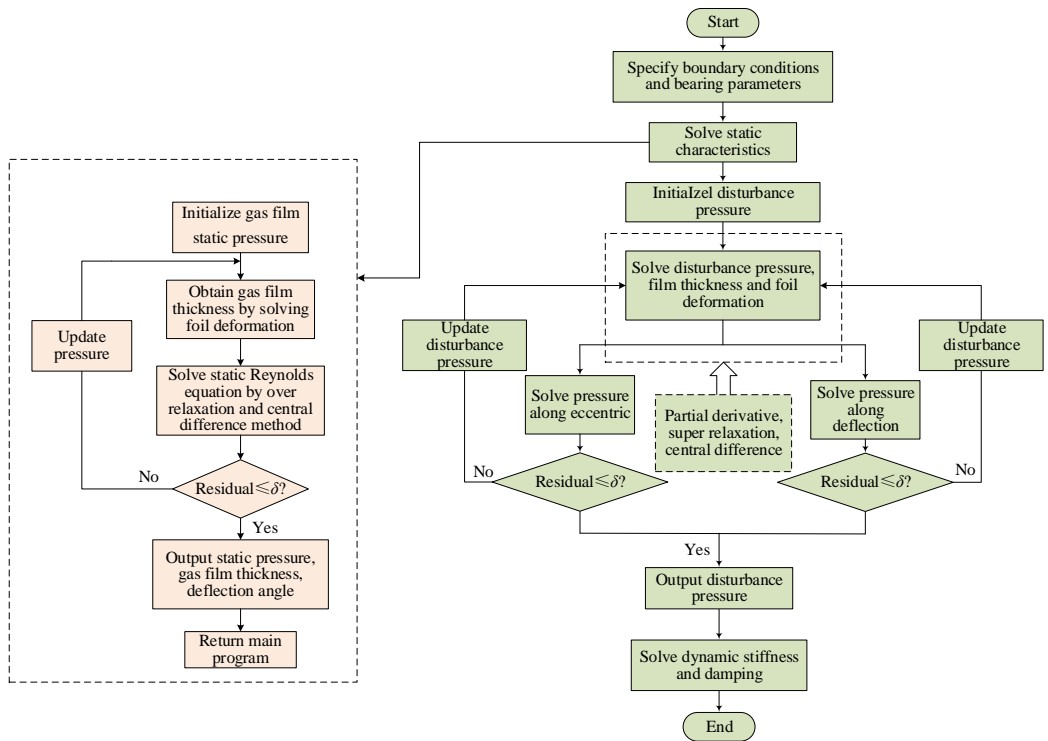

**Figure 5.** Procedure of solving dynamic characteristics.

The dimensionless deformation and dimensionless disturbed deformation of the foil can be expressed as follows:

$$\overline{Z}_0(i,j) = (\overline{p}_0(i,j) - 1)\overline{S}_1 / \overline{K}(i,j) \tag{10}$$

$$\overline{p}_d(i,j)\overline{S}_1 = \overline{K}_n(i,j)\left(\overline{Z}_d(i,j) + \eta\frac{\partial \overline{Z}_d(i,j)}{\partial \overline{t}}\right) \tag{11}$$

The position of the rotor in real-time can be expressed as follows:

$$\begin{cases} \varepsilon = \varepsilon_0 + E = \varepsilon_0 + E_a e^{i\,\overline{f}\,\overline{t}} \\ \varphi = \varphi_0 + \Theta = \varphi_0 + \Theta_a e^{i\,\overline{f}\,\overline{t}} \end{cases} \tag{12}$$

where $E_a$ is the amplitude of the disturbance eccentricity, $\Theta_a$ is the amplitude of the disturbance deflection angle, $\overline{f}$ is the dimensionless rotor disturbance frequency ($\overline{f} = v/\omega$), and $\overline{t}$ is a dimensionless time item. The pressure and thickness of a gas film can be expressed as follows:

$$\begin{cases} \overline{p}(i,j) = \overline{p}_0(i,j) + \overline{p}_{\mathrm{d}}(i,j) = \overline{p}_0(i,j) + \overline{p}_{\mathrm{da}}(i,j)e^{i\,\overline{f}\,\overline{t}} \\ \overline{h}(i,j) = \overline{h}_0(i,j) + \overline{h}_{\mathrm{d}}(i,j) = \overline{h}_0(i,j) + \overline{h}_{\mathrm{da}}(i,j)e^{i\,\overline{f}\,\overline{t}} \end{cases} \tag{13}$$

where $\overline{p}_0$ is the dimensionless static pressure of the gas film, $\overline{h}_0$ is the dimensionless static thickness of the gas film, $\overline{p}_{\mathrm{d}}$ is the dimensionless disturbed pressure of the gas film, $\overline{h}_{\mathrm{d}}$ is the dimensionless disturbed thickness of the gas film, $\overline{p}_{\mathrm{da}}$ is the amplitude of the dimensionless disturbed pressure of the gas film, and $\overline{h}_{\mathrm{da}}$ is the amplitude of the dimensionless disturbed thickness of the gas film. The Reynolds equation under disturbance can be expressed as:

$$\begin{aligned} &\frac{\partial}{\partial\theta}\left(\overline{p}_{\mathrm{da}}\overline{h}_0^3\frac{\partial\overline{p}_0}{\partial\theta}\right) + \frac{\partial}{\partial\lambda}\left(\overline{p}_{\mathrm{da}}\overline{h}_0^3\frac{\partial\overline{p}_0}{\partial\lambda}\right) + \frac{\partial}{\partial\theta}\left(3\overline{p}_0\overline{h}_0^2\overline{h}_{\mathrm{da}}\frac{\partial\overline{p}_0}{\partial\theta}\right) + \frac{\partial}{\partial\lambda}\left(3\overline{p}_0\overline{h}_0^2\overline{h}_{\mathrm{da}}\frac{\partial\overline{p}_0}{\partial\lambda}\right) \\ &+ \frac{\partial}{\partial\theta}\left(\overline{p}_0\overline{h}_0^3\frac{\partial\overline{p}_{\mathrm{da}}}{\partial\lambda}\right) + \frac{\partial}{\partial\lambda}\left(\overline{p}_0\overline{h}_0^3\frac{\partial\overline{p}_{\mathrm{da}}}{\partial\lambda}\right) = \Lambda\frac{\partial}{\partial\theta}\left(\overline{p}_0\overline{h}_{\mathrm{da}} + \overline{p}_{\mathrm{da}}\overline{h}_0\right) + i2\Lambda\overline{f}\left(\overline{p}_0\overline{h}_{\mathrm{da}} + \overline{p}_{\mathrm{da}}\overline{h}_0\right) \end{aligned} \tag{14}$$

It is noted that the static characteristic of the GDFBs, which indicates the performance in equilibrium state, is closely related to how such an equilibrium state is reached. Therefore, the initial state was specified as static in the fluid domain obtained in MATLAB. In contrast, the dynamic characteristic of the GDFBs indicates the sliding of the elastic foils with perturbation.

### 2.5. Validations of Computational Method

Currently, experimental measurements in which contact friction has been strengthened are greatly lacking. Although some researchers have obtained the aeroelastic characteristics of GDFBs, their details are still inadequate. Figure 6 illustrates the validation of the numerical method used in this study. The load capacity and clearance height obtained by the computational method agreed well with the experimental measurements obtained by Baum et al. [37] and Ruscitto et al. [38], respectively. For instance, at $\eta = 0.1$ and $\varepsilon = 0.3$, the curve of the load capacity in the numerical simulation almost resembled that obtained by Baum et al. [37]. With an increase in the eccentricity ratio of 0.7, the discrepancy in the pressure was within 10.2%, which was attributed to the GDFBs behaving close to the rigid case. In contrast, without friction, the error between the numerical simulation results and experimental data was as high as 16.6%, and the relative displacement between the top foil and the bump foil was highest compared with the case coupled with the contact friction. Compared with the simulation results under non-friction, those under contact friction were closer to the experimental data. Furthermore, the thickness distribution of the gas film was compared with that reported by Ruscitto et al. [38]. When there was no friction, the error between the numerical simulation results and experimental data was approximately 48.5%. In contrast, the error between the results of the two-way fluid–structure coupling numerical simulation and the experiments in this study was within 21.5%. This is attributed to the presence of the contact friction strengthening the stiffness, and the magnitude of deformation of the elastic foils being suppressed as the result. Discrepancies between the numerical simulations with BFSIs employed in this study and those in published works occurred due to two reasons. The first reason was the complicated operating conditions of the experimental measurement, and published works have not been presented in adequate detail. The second reason is the assumption of axial rotation without defection in the simulation, which is inconsistent with that in actual operation. In conclusion, numerical simulations with Bidirectional Fluid–Structure Interactions (BFSIs) can reveal the characteristics of bearings under bump foil deformation more accurately and reasonably.

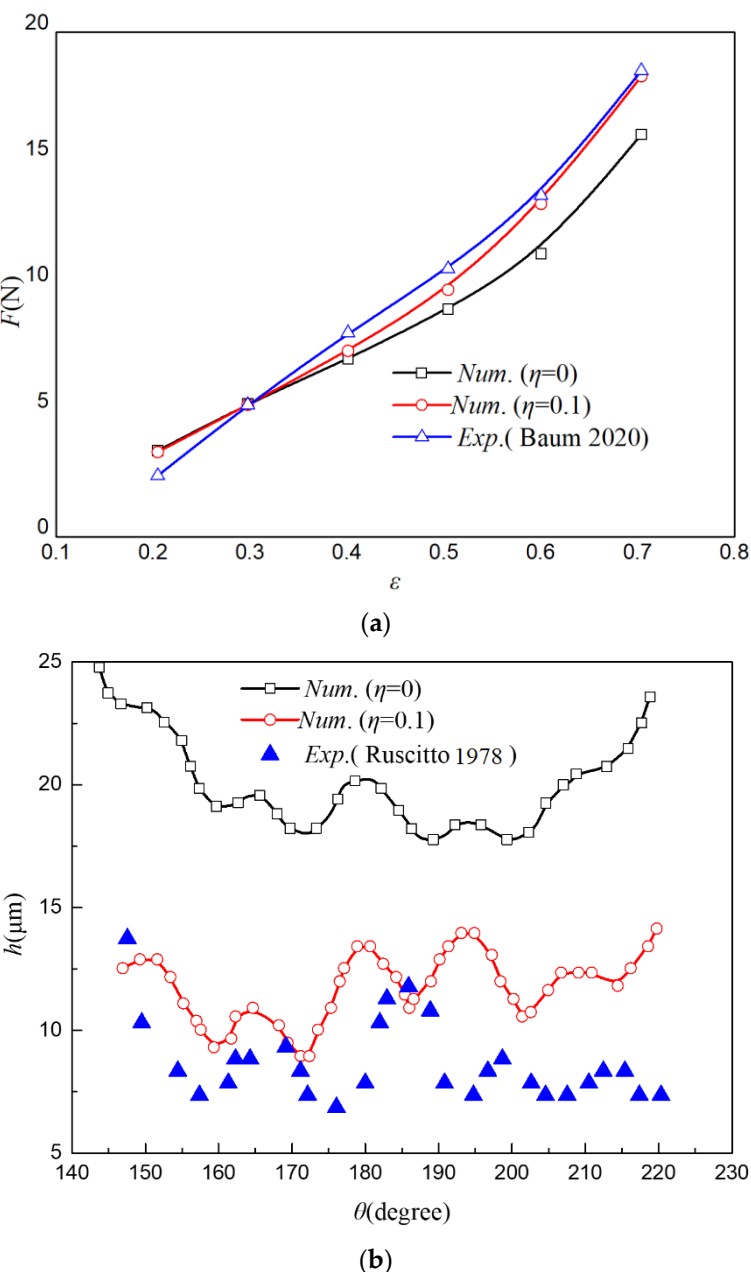

**Figure 6.** Validation of the numerical method in this study. (**a**) *F* vs. *ε* (Adapted with permission from Ref. [37]. 2020, Elsevier); (**b**) *h* vs. *θ* (Adapted with permission from Ref. [38]. 1978, Automobiles).

## 3. Results and Discussion

### 3.1. Static Characteristics

#### 3.1.1. Static Stiffness

This section discusses the static characteristics, including the stiffness, load capacity, and deflection angle. The top foil–top foil bearing, top foil–bump foil bearing, and the housing of the bump foil–bearing all had the same dynamic friction coefficients specified. Figure 7 shows the distribution of the nondimensional stiffness along the circumferential direction for different friction coefficients. The maximum nondimensional stiffness occurred near the midpoint between the key blocks, where the preload was employed. With the support of the bump foils and key blocks, it behaved like a rigid body. As it was far away from the key block, the static stiffness decreased sharply. For instance, at $\theta = 0.2\pi$, its static stiffness approached zero. As the friction coefficient increased, the static stiffness of the foil assembly increased, as shown in Figure 7a. Figure 7b shows the assembly of

the top and bump foil in the initial and deformed states at $\eta$ = 0.1. The dashed line plus non-filling indicates the initial state, while the solid line plus filling is the deformed state. The sliding motion was accompanied by the deformation of the elastic foils induced by aerodynamic force.

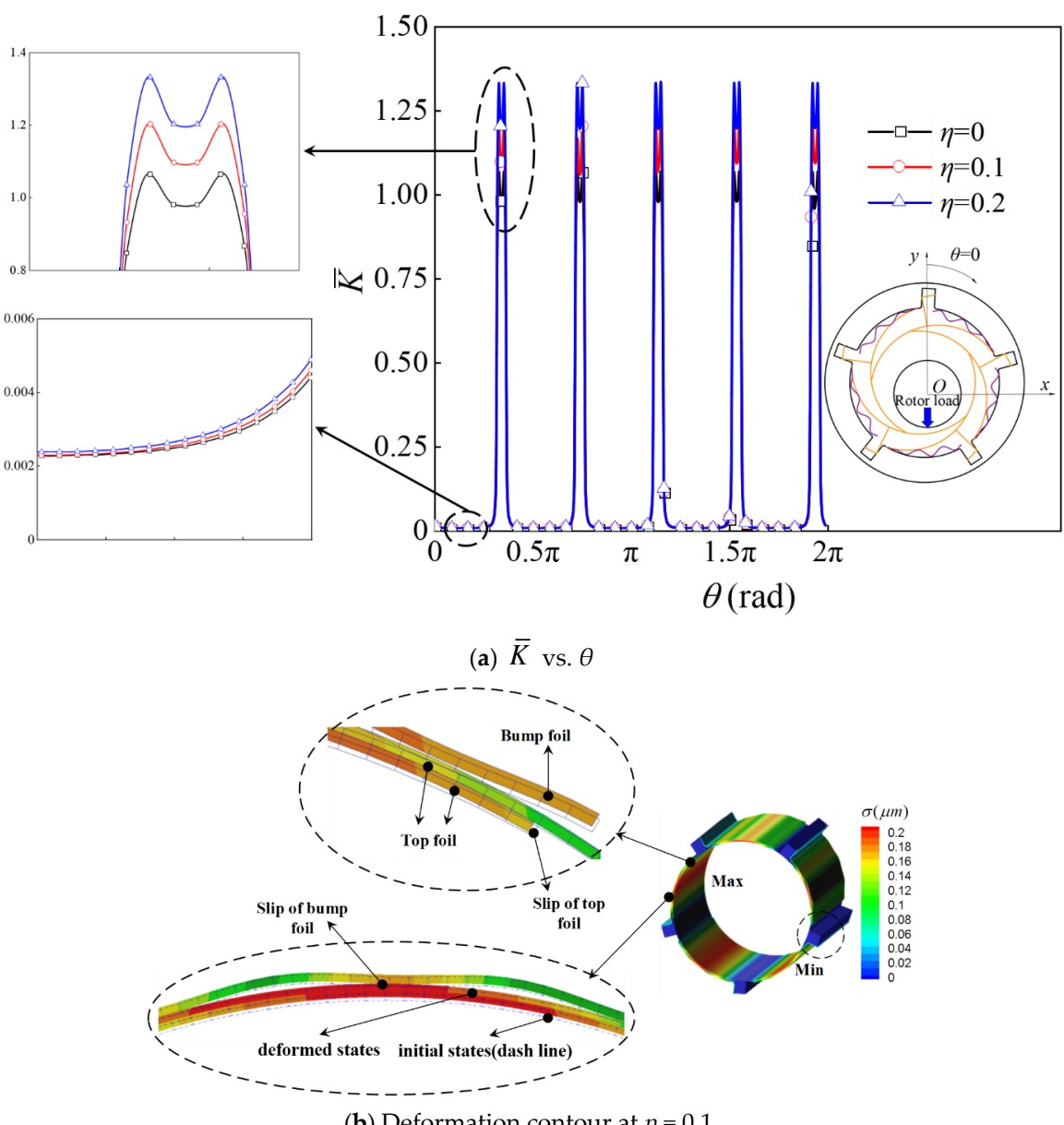

(a) $\bar{K}$ vs. $\theta$

(b) Deformation contour at $\eta$ = 0.1.

**Figure 7.** Distribution of nondimensional stiffness along circumferential direction with different friction coefficients.

In summary, the distribution of steady stiffness was closely associated with the configuration of the foil assembly and the state of contact friction. The maximum steady stiffness occurred near the key blocks and was accompanied by the support of the bump foils. As the magnitude of the load increased, the friction force sharply increased because the state of contact varied from line contact to face contact. Additionally, the magnitudes of both the frictional force and contact area also increased. Overall, the increase in the contact friction force strengthened the stiffness of the foil assembly, which was consistent with the experimental measurements [7,41].

### 3.1.2. Load Capacity and Deflection Angle

The load capacity and deflection angle at $\varepsilon = 0.5$ and $c = 40$ μm are presented and discussed here. Figure 8 shows the distribution of nondimensional pressure along the circumferential direction at $\Lambda = 0.75$. Figure 9 shows the radial deformation contours of the foil assembly at a bearing number of $\Lambda = 0.75$. As shown in Figure 8, near the region of negative pressure, the top foil deformed inward, and its pressure increased correspondingly. The eccentricity resulted in the same magnitude of deformation near the maximum-pressure region. Near the negative-pressure region, the magnitude of deformation decreased with an increase in the friction coefficient. Additionally, the nondimensional load capacity increased with an increase in the bearing number. The static stiffness increased with the friction coefficient; therefore, the positive pressure increased slightly as the friction coefficient increased. The dimensionless film thickness was close to that of the rigid bearing at $\eta = 0.1$, because the increase in the friction coefficient strengthened the stiffness of the foil assembly. Negative pressure existed near the region of $\theta = 1.25$–$1.5\pi$ and the existing contact friction hindered the pull-back of the top foils. As shown in Figure 9, for $\eta = 0$, the maximum and minimum deformation occurred at $\theta = 0.7$ and $1.5\pi$, respectively. As the friction coefficient increased, the magnitude of the deformation decreased. As the friction coefficient increased, the relative motion among the foils decreased because the frictional force increased. Consequently, the stiffness of the foil assembly increased, making it prone to rigidity. It is noted that the difference between the maximum and minimum pressure for $\eta = 0$ was larger than that in the cases with contact friction. This was because the relative motion between the top foil and the bump changed the distribution of the pressure of the gas film.

Overall, contact friction significantly affected the load capacity and deflection angle. Along the circumferential direction, the contact friction hindered the pull-back of the top foils near the negative pressure, impairing the load capacity of the GDFB. The normal load capacity and deflection angle initially increased and then remained unchanged as the coefficient increased. In contrast, the tangential load capacity first decreased slightly and then remained constant as the coefficient increased. The contact friction increased the stiffness of the foil assembly, which was unfavorable to its operational stability, and the GDFB was prone to rigidity.

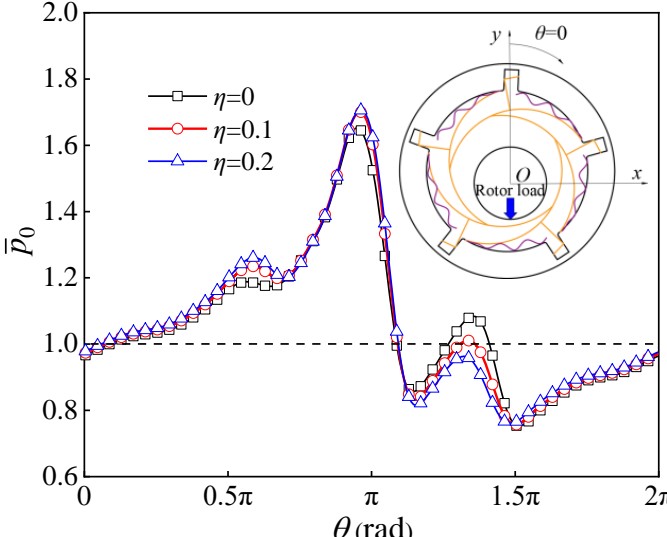

**Figure 8.** Distribution of nondimensional pressure along circumferential direction at $\Lambda = 0.75$.

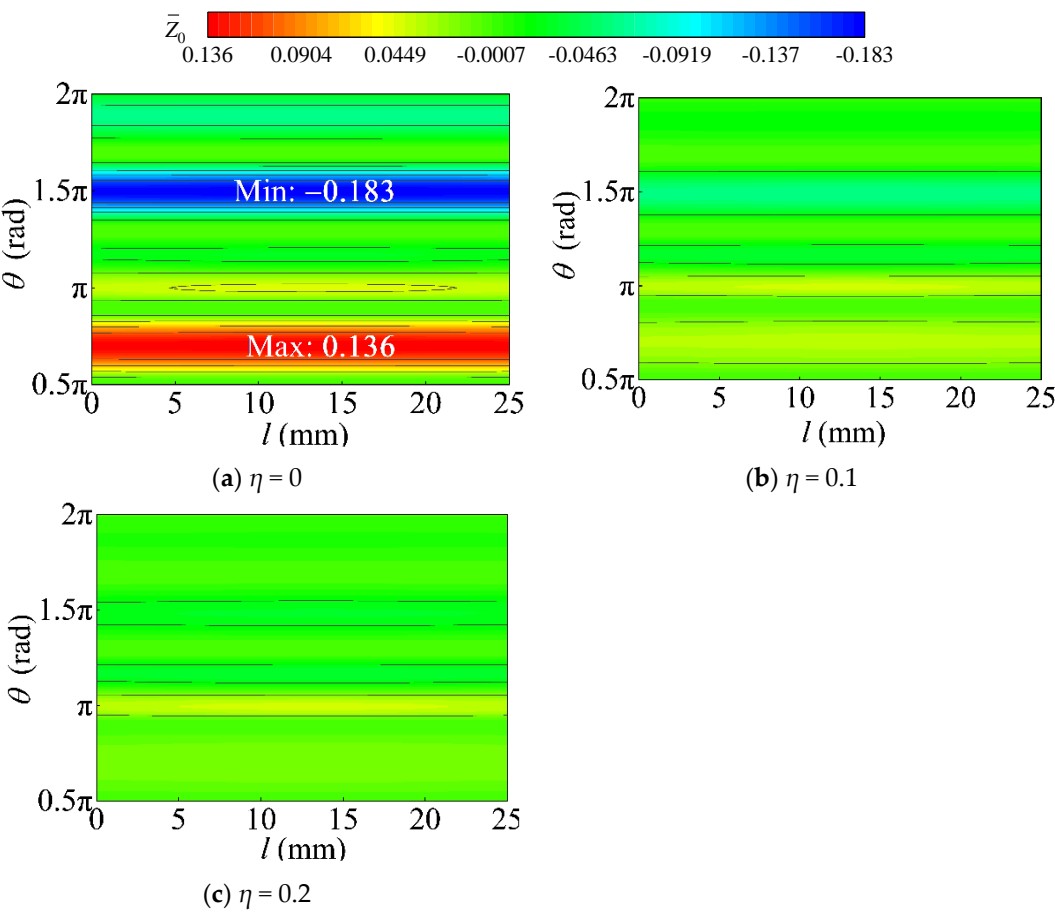

**Figure 9.** Radial deformation contours of foil assembly at $\Lambda$ = 0.75.

3.1.3. Dependent Factors

The influence of dependent factors, including the rotational speed and eccentricity, on the nondimensional load capacity and deflection angle was investigated. Figure 10 shows the variation in the nondimensional load capacity and deflection angle with rotational speed. Figure 11 shows the variation in the nondimensional pressure along the circumferential direction at the central section. Figure 12 shows the nondimensional pressure contours at different rotational speeds. Figure 13 shows the nondimensional radial deformation contours of the foils at different rotational speeds. As shown in Figure 10, as the rotational speed increased, the load capacity increased, whereas the deflection angle decreased. Furthermore, as the eccentricity ratio increased, the load capacity increased, whereas the deflection angle decreased. Contact friction only strengthened the load capacity when the rotational speed and eccentricity ratio exceeded $1.2 \times 10^5$ rpm and 0.3, respectively. Therefore, the effect of friction on the load capacity was more sensitive to large eccentricity ratios and rotational speeds. For instance, at a small eccentricity ratio of $\varepsilon$ = 0.2, the load capacity at $\eta$ = 0.1 almost resembled that at $\eta$ = 0, while the load capacity at $\eta$ = 0.1 was 39.5% larger than that at t $\eta$ = 0. As shown in Figure 11, as the rotational speed increased, the maximum pressure increased, and the presence of contact friction further intensified this increase. The pressure became negative near the region of $\theta \in (1.25\pi, 1.5\pi)$, while the pressure was larger than zero because of the pull-back of the top foils when $\eta$ = 0. Figure 12 shows the nondimensional pressure contours at different rotational speeds. The contact friction changed the distribution of the gas film pressure, inhibiting the pull-back of the top foils. The maximum pressure of the gas film was inconsistent with the maximum deformation along the circumferential direction because of the configuration of the foil assembly. The top foil deformed toward the rotator, the thickness of the gas film decreased, and the pressure increased. As shown in Figure 13, the maximum nondimensional radial

deformation at $\eta = 0$ was larger than at $\eta = 0.1$ near $\theta = 0.6\pi$, while the minimum radial deformation at $\eta = 0$ was less than that at $\eta = 0.1$ near $\theta = 1.4\pi$. This was attributed to the presence of elasticity of the foil flattening the distribution of the pressure of the gas film. The magnitude of the deformation weakened as a result.

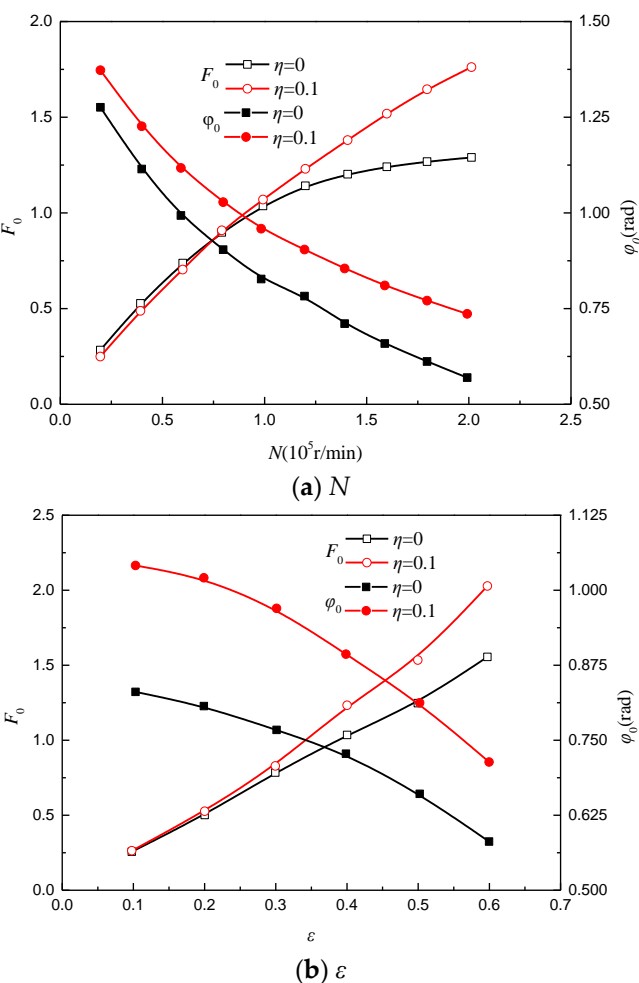

**Figure 10.** Variation in nondimensional load capacity and deflection angle with rotational speed.

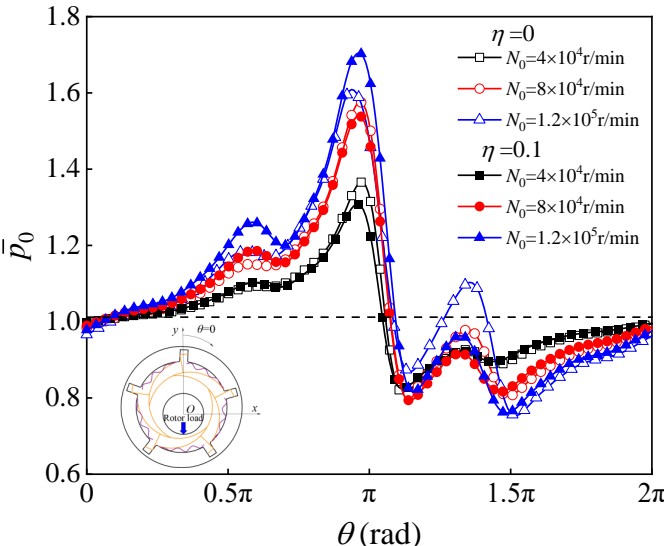

**Figure 11.** Variation in nondimensional pressure along circumferential direction at central section.

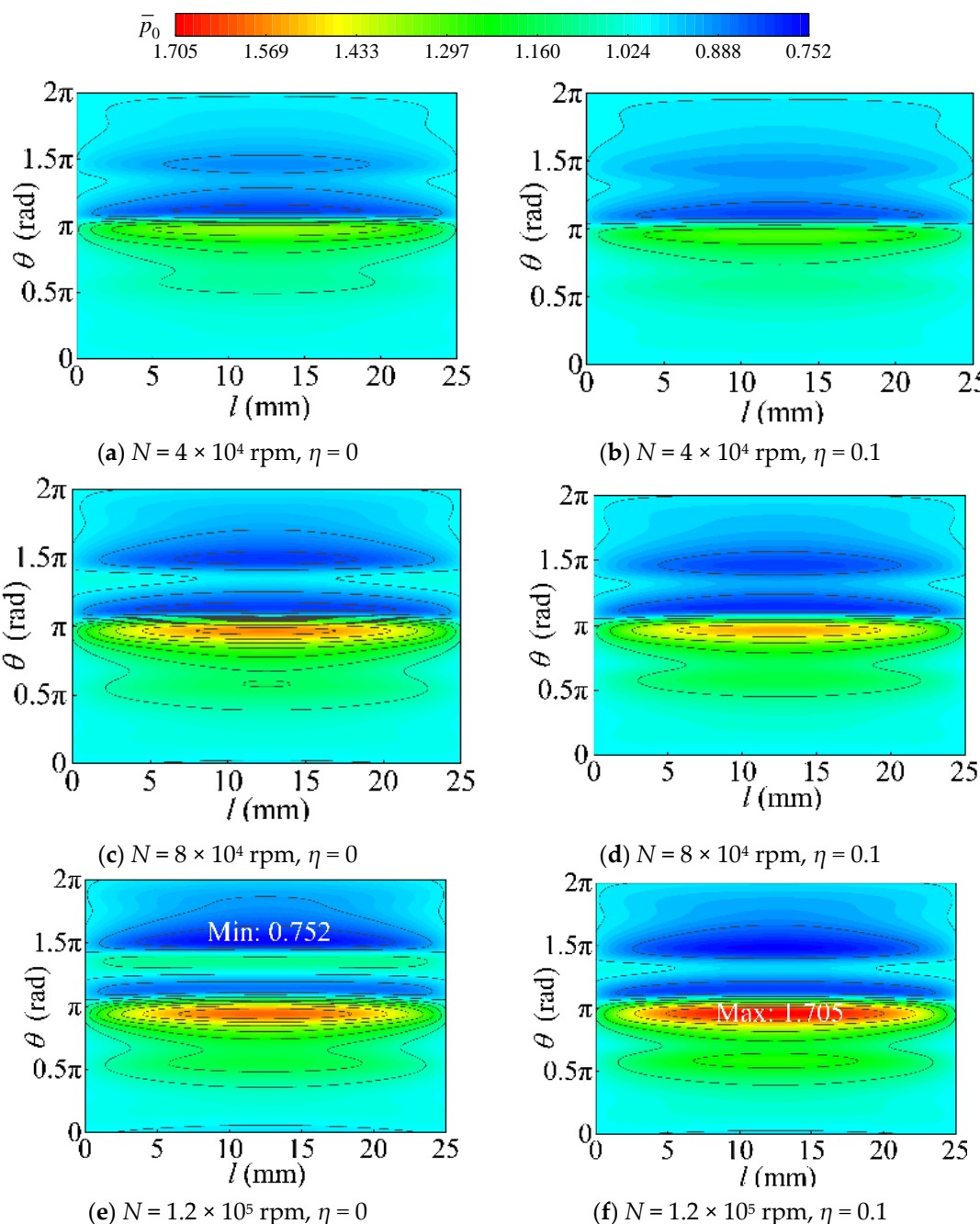

**Figure 12.** Nondimensional pressure contours at different rotational speeds.

To summarize, the effects of rotational speed and eccentricity on the nondimensional load capacity and deflection angle were highly associated with the contact friction. The nondimensional load capacity first increased with an increase in the friction coefficient. Subsequently, it remained unchanged as the friction coefficient continued to increase, which was consistent with the observations of previous researchers [18,19]. The increase in the rotational speed and eccentricity enhanced the load capacity while weakening its operational stability. Contact friction could only strengthen the load capacity when the rotational speed and eccentricity ratio exceeded $1.2 \times 10^5$ rpm and 0.3, respectively. The maximum pressure of the gas film was inconsistent with the maximum deformation along the circumferential direction because of the configuration of the foil assembly.

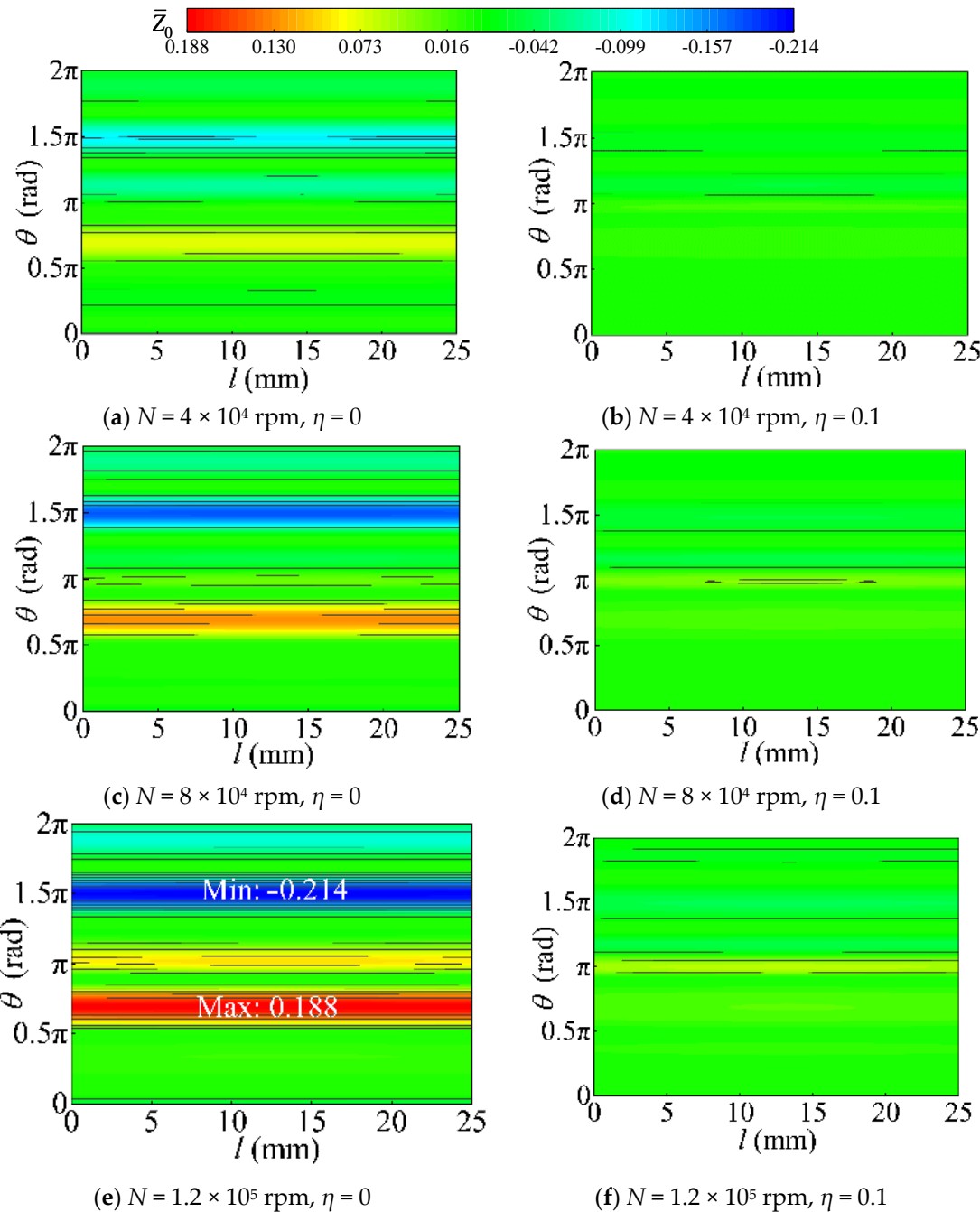

**Figure 13.** Nondimensional radial deformation contours of foils at different rotational speeds.

*3.2. Dynamic Response*

This study presents the characteristics of the dynamic response of a GDFB. Figures 14 and 15 show the variations in the dynamic stiffness and dynamic damping with the nondimensional disturbance frequency. Here, the dynamic stiffness took place when the GDFB was under dynamic incentive (perturbation), while the static stiffness indicated the variation in stress with strain. Additionally, the damping could be used to describe the energy dissipation of the GDFB. As shown in Figure 14, as the nondimensional disturbance frequency increased, the primary dynamic stiffnesses $k_{xx}$ and $k_{yy}$ increased, while the cross-dynamic stiffnesses $k_{xy}$ and $k_{yx}$ first increased and decreased, respectively, and then remain unchanged, maintaining consistency with the experimental measurements of ref. [42]. The dynamic stiffness increased with an increase in the friction coefficient. In this study, the key block came into contact with the slot of the bearing housing with the contact constraint. The

fixed constraint yielded a larger dynamic stiffness than the contact constraint employed in this study. As shown in Figure 15, overall, the dynamic damping $d_{xx}$, $d_{yy}$, and $d_{xy}$ decreased, while the dynamic cross-damping $d_{yx}$ decreased as the nondimensional disturbance frequency increased; then, $d_{xx}$ decreased with an increase in the friction coefficient at a small nondimensional disturbance frequency, whereas $d_{xx}$ remained unchanged at a large nondimensional disturbance frequency. Obviously, the fixed constraint resulted in a larger dynamic stiffness and less dynamic damping, particularly at a small nondimensional disturbance frequency, which did not support its operational stability. The dynamic damping decreased with an increase in the nondimensional disturbance frequency, which was attributed to the denseness of the gas film decreasing the energy dissipation.

Overall, the contact friction prevented the relative motion between the top and bump foils, consequently strengthening the dynamic stiffness. On one hand, the presence of contact friction induced additional energy dissipation. On the other hand, the integrated stiffness increased correspondingly, and the energy dissipation weakened. Compared with the contact constraint, the fixed constraint of the key block increased the dynamic cross stiffness, which was unfavorable for operational stability, and the dynamic cross-damping $d_{yx}$ increased. This showed the extremely complicated nature of the constrained state-coupled contact friction of the key block, and it requires further exploration.

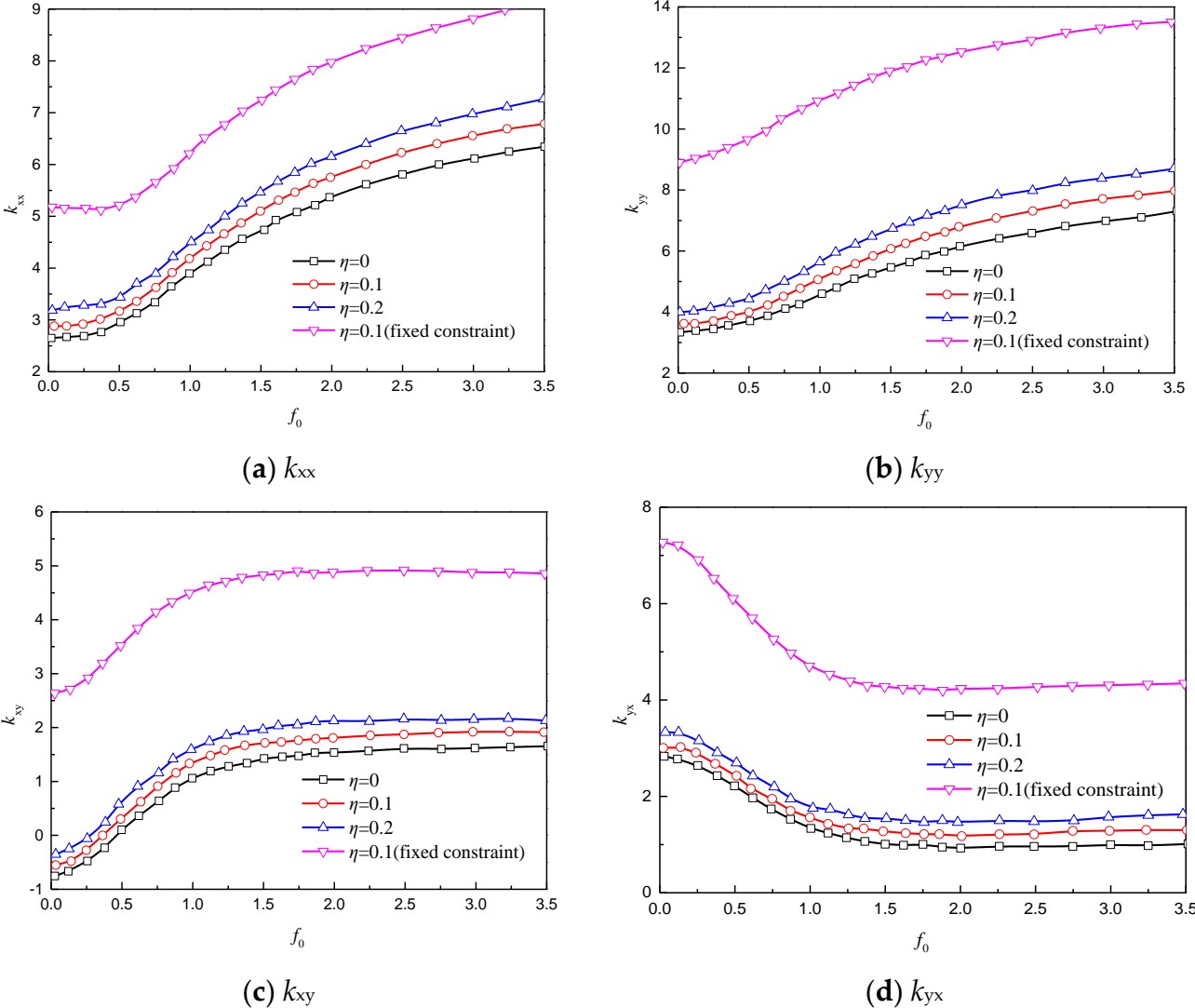

**Figure 14.** Variation in dynamic stiffness with nondimensional frequency.

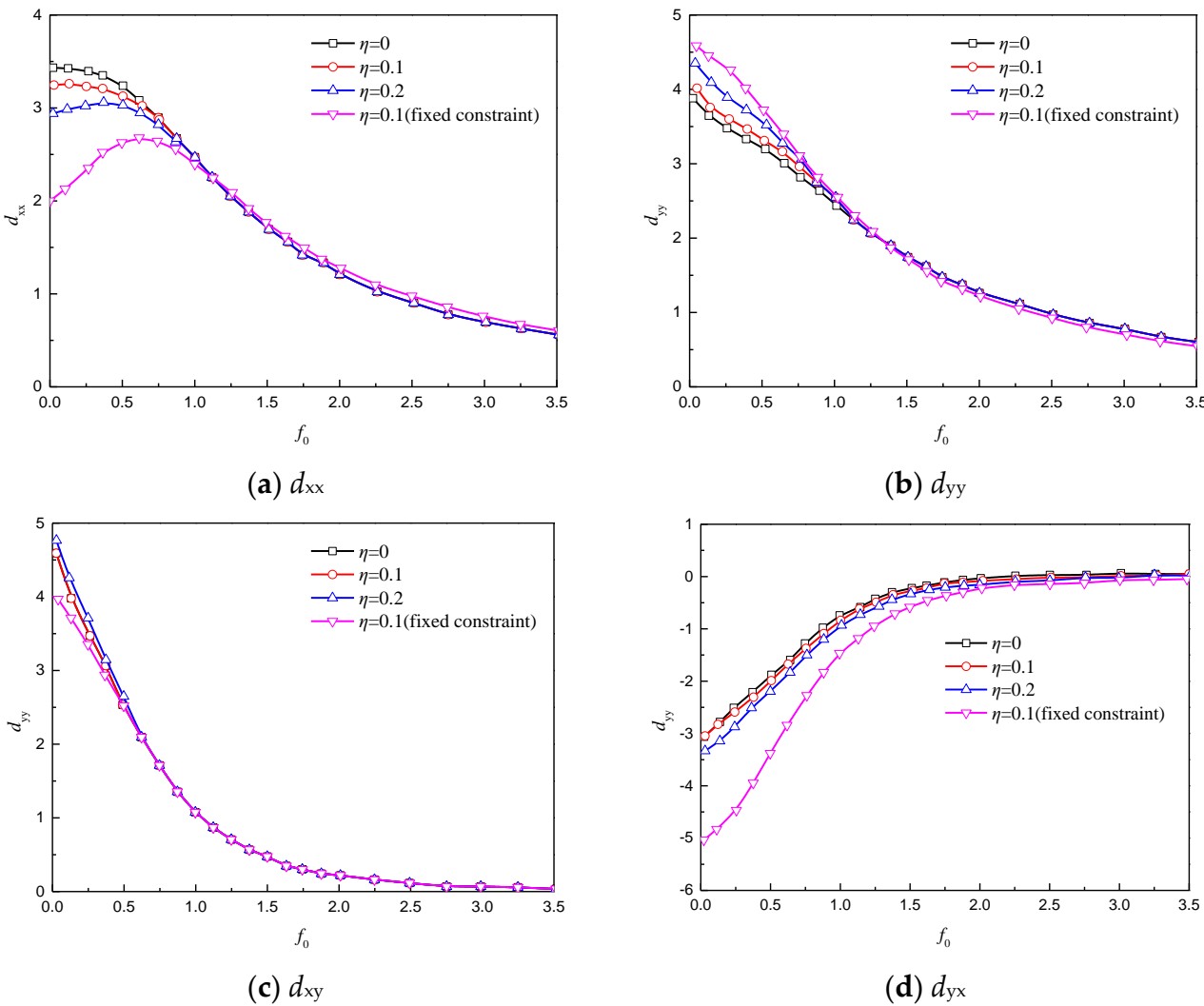

**Figure 15.** Variation in dynamic damping with nondimensional frequency.

## 4. Conclusions

In this study, a numerical method that considered BFSIs was developed. The effects of the contact friction and deformation of the elastic foils on the static and dynamic characteristics of the GDFB were explored, and the results are presented here.

(1) The distribution of the steady stiffness was closely associated with the configuration of the foil assembly and the state of contact friction. An increase in the contact friction force increased the stiffness of the foil assembly, wherein the state of contact varied from line contact to face contact. The maximum steady stiffness was near the key blocks and was accompanied by the support of the bump foils. The load capacity increased with the friction coefficient. The presence of contact friction hindered the pull-back of the top foils near the negative pressure, which impaired the load capacity of the GDFB. The normal load capacity and deflection angle initially increased and then remained unchanged as the coefficient increased. The contact friction increased the stiffness of the foil assembly, which was unfavorable for its operational stability, and the GDFB was prone to rigidity.

(2) The effects of the rotational speed and eccentricity on the nondimensional load capacity and deflection angle were highly associated with contact friction. Overall, the increase in the rotational speed and eccentricity enhanced the load capacity while weakening the operational stability. The presence of contact friction could only strengthen the load capacity when the rotational speed and eccentricity ratio exceeded $1.2 \times 10^5$ rpm and 0.3, respectively. The maximum pressure of the gas film was inconsistent with the

maximum deformation along the circumferential direction because of the configuration of the foil assembly.

(3) The contact friction prevented the relative motion between the top foil and bump foil bearings. Consequently, the dynamic stiffness was strengthened. The contact friction induced additional energy dissipation, whereas increases in the integrated stiffness inhibited its energy dissipation. Compared with the contact constraint, the fixed constraint of the key block increased the dynamic cross-stiffness, which was unfavorable for operational stability. The constrained state-coupled contact friction of the key block was extremely complicated; thus, it requires further exploration.

**Author Contributions:** Conceptualization, C.Y., Y.L. and J.Z.; methodology, B.X., Z.W. and Y.L.; software, C.Y. and B.X.; validation, C.Y., B.X. and Z.W.; formal analysis, C.Y. and B.X.; investigation, C.Y. and Z.W.; resources, Y.L.; data curation, B.X.; writing—original draft preparation, C.Y.; writing—review and editing, Z.W., Y.L., B.X. and J.Z.; visualization, C.Y.; supervision, J.Z.; project administration, J.Z.; funding acquisition, Y.L. and J.Z. All authors have read and agreed to the published version of the manuscript.

**Funding:** This research was funded by the Aeronautical Science Foundation of China (Grant No. 201928052008), National Natural Science Foundation of China (Grant No. 52206091), Natural Science Foundation of Jiangsu Province (Grant No. BK20210303), and Advanced Jet Propulsion Innovation (Grant No. HKCX2022-01-001).

**Institutional Review Board Statement:** Not applicable.

**Informed Consent Statement:** Not applicable.

**Data Availability Statement:** Not applicable.

**Conflicts of Interest:** The authors declare no conflict of interest.

## Nomenclature

| | |
|---|---|
| $\Lambda$ | Bearing number $\Lambda = 6\mu\omega R^2/(P_a c^2)$ |
| $h$ | Thickness of the gas film (m) |
| $\varepsilon$ | Journal eccentricity |
| $\theta$ | Angular coordinates |
| $h_z$ | Radial deformation of the inner surface (m) |
| $\overline{Z}(i,j)$ | Radial dimensionless displacement |
| $\overline{G}_0(i,j)$ | Dimensionless distance between the top foil and rotator |
| $\bar{t}$ | Dimensionless time $\bar{t} = vt$ |
| $p$ | Pressure (Pa) |
| $H$ | Dimensionless clearance height $H = h/c$ |
| $R$ | Journal radius (m) |
| $Pa$ | Environmental pressure (Pa) |
| $c$ | Nominal bearing clearance |
| $K$ | Nominal static stiffness |
| $\eta$ | Coefficient of friction |
| $K_n$ | Static stiffness of $K(i,j)$ |
| $v$ | Disturbance frequency (Hz) |
| $E_a$ | amplitude of disturbance eccentricity |
| $\Theta_a$ | Amplitude of disturbance deflection angle |
| $\overline{f}$ | Dimensionless rotor disturbance frequency $(\overline{f} = v/\omega)$ |
| $\bar{t}$ | Dimensionless time |
| $\overline{p}_0$ | Dimensionless static pressure |
| $\overline{h}_0$ | Dimensionless static thickness |
| $\overline{p}_d$ | Dimensionless disturbed pressure |
| $\overline{h}_d$ | Dimensionless disturbed thickness |
| $\overline{p}_{da}$ | Amplitude of the dimensionless disturbed pressure |
| $\overline{h}_{da}$ | Amplitude of the dimensionless disturbed thickness |

Greek Letters
$\rho$      Density (kg/m$^3$)
$\theta$      Angular coordinate
$\lambda$      Axial coordinate
$\beta$      Top foil opening angle
$\mu$      Dynamic viscosity (Pa·s)
$\mu_t$      Turbulent dynamic viscosity (Pa·s)
Subscripts
$b$      Bump
$f$      Fluid
$s$      Solid
$t$      Top

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
