# Peer review of "Numerical Investigation of Gas Dynamic Foil Bearings Conjugated with Contact Friction by Elastic Multi-Leaf Foils"

_aerospace, doi:10.3390/aerospace10070585_

Round 1

Reviewer 1 Report

The review is about the numerical investigation of gas dynamic foil bearings conjugated with contact friction by elastic multi-leaf foils. It is interesting and may be used as a suture reference, to make it possible it needs to be corrected in these aspects:

1.           In Figure 2 (a) is a description, that the solver of ABAQUS was nonlinear. What was the source of the nonlinearity?

2.           Please define the global coordinate system. How data, for example, from Figure 7 should be correlated with it?

3.           Figure 9 shows a big separation between the min results, and the max results, please comment on why is such a separation.

4.           Authors wrote: The static stiffness of the bearings did not change linearly with speed but varied proportionally with the elastic modulus. Could authors explain what it exactly means?

5.           Authors wrote: A calculation formula to solve the variable stiffness problem of bump foils has been derived from an experiment on the stiffness of bearing structures. Please add a source to the appropriate paper.

6.           It is better to write rpm instead of r/min.

7.           Authors wrote that the dynamic stiffness increases with the bearing number and decreases with increasing bearing compliance. Please add more details to this one and add an appropriate reference.

8.           There is a sentence Figure 2 shows the… it should be, Figure 2 (a).

9.           Add spaces between the figures and parts a and b. Instead, Figure 3(b) should be Figure 3(b). This should be unified across drawings and in the text.

10.         Please add the values of parameters from equations 8 and 9 for the nominal conditions.

11.         Description of Figure 5 is about Figure 6.

12.         It would be good to expand the introduction section. These papers may be helpful for the authors: 10.1007/s40544-020-0427-7,  https://doi.org/10.3390/app11020878, 10.1016/j.ijmecsci.2022.107091.

Author Response

Thanks for your thoughtful and valuable comments. Your comments and suggestions help us to improve the quality of our work. The revised manuscript and Responses to Reviewer Comments had been upload the attachment.

Reviewer 2 Report

The article presents interesting research on the impact of friction on the properties of foil bearings. The authors focus on the analysis of various parameters, such as stiffness, pressure, and deformation, in the context of different friction coefficients. However, the article has several areas that could be improved to increase its clarity and usefulness to readers.

1. The lack of reference to Figure 6 in the text is problematic, as readers may have difficulty understanding what the figure is intended to illustrate.

2. It is not clear in the text what the friction coefficient η refers to. The authors should explain between which elements this friction occurs.

3. All notations included in the equations should be explained. This will help readers understand what each notation represents.

4. Figure 7 lacks an explanation of what the large letters A, B, C, D, and E mean. The authors should explain what these letters represent, and also answer the question of whether the stiffness values for each "key block" are identical for a given friction coefficient (Fig.7).

5. The charts are difficult to interpret because there is no description of the values shown on the axes. Readers have to search the text for what some notations mean.

6. The quality of Figure 11 should be improved so that the quality of all figures is identical.

7. Captions under the figures should be more descriptive and contain more information. This will help readers understand what each figure is intended to illustrate.

8. On the figures where the author shows the pressure distribution or deformations, the boundaries of the foils should be marked. In its current form, it is difficult to understand on which foil the lubricating film is formed or which foil is deformed.

9. The introduction lacks information on why this research is important. Why they are investigating these relationships. The authors should explain, for example, why the deflection angle is so important.

10. The work lacks information about the boundary conditions and contact settings in ABAQUS.

11. Figures should be pasted into the text because searching for them at the end of the article and comparing them with the text is tiring for the reader.

(In my opinion, the article could benefit from a better explanation of the purpose of the research. The authors should explain why they are investigating the impact of friction on dynamic gas foil bearings and why the deflection angle is so important. In addition, the article could benefit from better organization and formatting, including placing figures in the text, rather than at the end of the document.)

In summary, the article presents interesting and potentially important research, but could benefit from a more detailed and clear presentation of information. The authors should focus on improving the clarity of the charts, providing more details about their methodology and results, and better explaining the significance and implications of their research.

Author Response

Thanks for your thoughtful and valuable comments. Your comments and suggestions help us to improve the quality of our work. The revised manuscript and Responses to Reviewer Comments had been upload the attachment

Reviewer 3 Report

The authors study numerically static and dynamic properties of gas dynamic foil bearings with elastic multi-leaf foils considering the contact friction between the bearing elements. The topic of the study is actual and interesting for readers. The numerical results are compared with the experimental data from other authors and show the acceptable agreement, taking into account the complexity of interactions in a gas foil bearing. However, the article still needs some improvements, the comments and questions are below.

1. The introduction must be more structured and describe one after another individual background issues that are relevant in the context of this work. In the present form this section looks pretty chaotic.

2. Line 97, please explain, what do you mean by the term 'bearing number'?

3. The numerical procedures used for solving the problems shown in Fig.2 and Fig.4 require more clear and detailed explanation, including the used methods and their parameters, the assumptions made.

4. Given the complexity of the model and the iterative nature of the calculations, it is also necessary to provide the data on the actual tolerances used in the iterative calculations. It is also useful to evaluate the performance of the model and the required time to perform the calculations, indicating the hardware used.

5. How equations (8) and (9) have been obtained? And why the stiffness in (8) is in dimensionless form, while the damping in (9) is not?

6. Line 227, there should be Fig. 6 instead of 5.

7. Please explain in more detail how the discrepancies in percent in Section 2.5 have been calculated from pressure distributions.

8. According to a number of studies on gas foil bearings, stick-slip motion in dry friction contacts can affect the bearing operation significantly. This phenomenon should be reflected in the introduction, and in the model description, including the reasons for omitting it and the possible consequences of such simplification.

9. It would be useful to supplement Fig. 8 with a gap profile in the initial and deformed states in order to more clearly assess the deformations of the foils together with the pressure distribution.

10. Line 327, there should be Fig. 14 and 15 instead of 18 and 19.

11. It is advisable to explain in the discussion the relationship between the static and dynamic bearing stiffness.

12. It is also advisable to provide an estimation of the expected changes in energy losses mentioned in Lines 346-347. Since this statement is also made in the conclusion in lines 372-373, it must be supported by some calculations and analysis.

13. A significant claim to the work as a whole is the quality of English, it should be significantly improved.

Author Response

(The authors gave the same response as above.)

Reviewer 4 Report

In the paper the authors investigated the effect of friction between foils in gas bearings.

  1. Use the number [1] for citation instead of using the author name and year of paper
  2. In the first sentence of the abstract, the direct object (gas bearing) must be introduced
  3. Line 97: bearing number should be “foil number”?
  4. Line99: remove citation [40] and correspondin text because it refers to water hydrostatic bearing, that are far from gas bearing
  5. In the introduction please specify that the contact friction occurred between foils.
  6. Sentence from line 171 to 174 are unclear
  7. Section 2.3: again specify where friction contact occour. Maybe add an arrow in figure 1
  8. Add a nomenclature
  9. Line 199: LAMBDA is the number of bearings, should be “number of foils”?
  10. Line 227: figure 5 should be figure 6
  11. If the effect of friction it is clear in case of perturbation, it is not clear in static condition in section 3.1.1.   In static conditions, the tangential friction force T and its direction, depends on the equilibrium of forces. The limit of static condition of friction is given by |T|<mu_static*N, that is different by the case of sliding in which |T|=mu_dynamic*N. So it depends on how such equilibrium condition is reached, for example by increasing the speed or reducing the speed. For example by considering the time-history different equilibrium deflection condition can be reached.

At lines 246-247 the same friction coefficient is used on all the contact. Which friction coefficient? Static or dynamic? Please discuss this point

  1. Line 269 and caption of figure 9: radical should be “RADIAL”?
  2. Figure 7: please zoom also one area at high K

Author Response

(The authors gave the same response as above.)

Round 2

Reviewer 1 Report

Nearly all of my recommendations have been implemented, greatly enhancing the clarity of the article for future readers. However, I noted that the authors incorporated only two out of the three suggested articles. If feasible, I would appreciate the addition of the third recommended piece. The article, after this small correction, is excellently composed and I strongly recommend it for publication.

Author Response

Thanks for your thoughtful and valuable comments. Your comments and suggestions help us to improve the quality of our work. Based on your suggestion, we have added the third reference, shown as follows,

GU et al.[22-23] had adopted the LuGre dynamic friction model to capture accurate stick-slip states of the gas foil bearing(GFBs) which coupled with electrohydrodynamic and nonlinear contact friction. The results shows that Friction has an impact on linear and nonlinear stability of the GFBs. The energy dissipation induced by the foil structure is highly associated with the stick-slip states, such that the optimal friction coefficient considering its nonlinear stability is bigger than the linear stability. Brenkacz L et al.[24] employed the ultra-high-speed camera to measure the vibrations of elastic foils. it is found that camera can provided several vibrant pictures of the deformation of the elastic foils in gas foil bearings. The movement of the foils in the frequency domain can be analyzed and compared with numerical simulation.

Zhou R , Gu Y, Ren G. Modeling and stability characteristics of bump-type gas foil bearing rotor systems considering stick-slip friction[J]. International Journal of Mechanical Sciences, 2022(219-):219.

Yongpeng G U, Lan X , Ren G. An efficient three-dimensional foil structure model for bump-type gas foil bearings considering friction[J]. FRICTION, 2021(006):009.

Brenkacz L, Baginski P, Zywica G. Experimental Research on Foil Vibrations in a Gas Foil Bearing Carried Out Using an Ultra-High-Speed Camera[J]. APPLIED SCIENCES-BASEL, 2021, 11(2): 564-577.

Reviewer 3 Report

The authors provided complete and detailed answers to the questions, and made the necessary changes to the manuscript. I recommend to accept the paper in the present form.

Author Response

Thanks for your thoughtful and valuable comments. Your comments and suggestions help us to improve the quality of our work.